# Handheld snapshot multi-spectral camera at tens-of-megapixel resolution

Weihang Zhang [1,6], Jinli Suo [1,2,3,6] ✉, Kaiming Dong [1], Lianglong Li[1], Xin Yuan[4], Chengquan Pei[5] & Qionghai Dai [1,2] ✉

Multi-spectral imaging is a fundamental tool characterizing the constituent energy of scene radiation. However, current multi-spectral video cameras cannot scale up beyond megapixel resolution due to optical constraints and the complexity of the reconstruction algorithms. To circumvent the above issues, we propose a tens-of-megapixel handheld multi-spectral videography approach (THETA), with a proof-of-concept camera achieving 65-megapixel videography of 12 wavebands within visible light range. The high performance is brought by multiple designs: We propose an imaging scheme to fabricate a thin mask for encoding spatio-spectral data using a conventional film camera. Afterwards, a fiber optic plate is introduced for building a compact prototype supporting pixel-wise encoding with a large space-bandwidth product. Finally, a deep-network-based algorithm is adopted for large-scale multi-spectral data decoding, with the coding pattern specially designed to facilitate efficient coarse-to-fine model training. Experimentally, we demonstrate THETA's advantageous and wide applications in outdoor imaging of large macroscopic scenes.

Multi-spectral imaging captures the spectral profile of the target scene, to quantitatively characterize the constituent energy/colors of the material's radiation, and serves as one of the fundamental tools for material identification due to its wavelength-dependent absorption. Therefore multi-spectral imaging, especially with large throughput and a compact design, plays crucial roles in a broad range of applications, such as agriculture, pollution monitoring, gene sequencing, astronomy, etc[1]. Technically, multi-spectral imaging is far from being as straightforward as RGB imaging, since the Bayer pattern based mosaicing strategy can not be scaled up to tens or more color channels. Researchers have explored various ways to capture multi-spectral data (a three-dimensional spatio-temporal data cube) and made big progress in the past decades.

The primary studies focus on measuring the spectral profile of a single point and the earliest spectrometers enhancing spectral resolution may date back to the middle of the last century[2]. Later, many improved spectrometers have been proposed, built on filter arrays[3], quantum point[4] and metasurface[5]. An intuitive way of extending such single-point spectral recording to two-dimensional scenes is temporal scanning, via either collecting the spectrum of each scene point in turn[6,7] or capturing the scene appearance at equivalent spectrum intervals[8,9] sequentially using tunable filters[10–12], but is too slow for dynamic scenes.

To address this limitation, researchers have developed snapshot spectral imaging techniques[13] to capture the spatio-spectral cube within a single image. One representative solution is to compromise spatial resolution for spectral discrimination, using "super-pixels" to record different narrow-band spectra of a scene point collectively[14,15]. However, direct compromise between spatial and spectral resolution is limited in scenarios demanding both spatial details and spectral precision. Introducing an additional high-resolution RGB camera to build a hybrid imaging system can improve the spectral resolution[16–22], but faces challenges in cross-resolution registration and fidelity degradation as spectral resolution increases.

For high resolution multi-spectral imaging of dynamic scenes, researchers utilize the redundancy in nature scenes to encode the

[1]Department of Automation, Tsinghua University, Beijing 100084, China. [2]Institute of Brain and Cognitive Sciences, Tsinghua University, Beijing 10008, China. [3]Shanghai Artificial Intelligence Laboratory, Shanghai 200232, China. [4]WestLake University, Hangzhou 310030 Zhejiang, China. [5]Xidian University, Xi'an 710071 Shaanxi, China. [6]These authors contributed equally: Weihang Zhang, Jinli Suo. ✉e-mail: jlsuo@tsinghua.edu.cn; qhdai@tsinghua.edu.cn

spatio-spectral data cube into a snapshot compactly, and decode computationally afterwards[23]. Researchers have designed various encoded snapshot spectral imaging setups, in which disperser[24-26] or diffuser[27,28] play important roles and emerging metasurface further promotes the miniaturization[29,30]. Among these encoding schemes, the compressive sensing method enables high spatio-spectral acquisition and inspires some new setups, including the precedent Coded Aperture Snapshot Spectral Imager (CASSI)[31] and its variants for performance improvement or system compactness[20,32-50]. Besides, further introducing temporal continuity shares the same mathematical model as CASSI[51] and can achieve multi-spectral videography with increased frame rate[52]. With an accurate transmission model, the aforementioned tunable liquid crystal filter[53,54] also serves as a controllable spectral coding approach and integrates information from hundreds of wavebands into a few acquisitions, thus can be combined with the coded aperture to achieve joint compression in both spatial and spectral domains[55,56]. This scheme demonstrates improved quality of multi-spectral reconstruction with several advantages, e.g., ease of use, high image quality, and light weight despite the need for multiple measurements, which may limit its applications in dynamic scenes.

For a comprehensive review, please refer to the article by ref.[57] As for compressive-sensing-based decoding, since multi-spectral data reconstruction from a snapshot is an under-determined problem, most algorithms incorporate priors into an optimization framework or a deep neural network. The widely used optimization framework include generalized alternating projection (GAP)[58], alternating direction method of multiplier (ADMM)[59], etc. The diverse prior models can be generally classified into pixel level[60,61], patch level[20,62,63], non-local similarity[21,64], cross channel similarity[22] or deep priors learned by a deep neural network[65-70]. To improve the reconstruction efficiency further, researchers try to build end-to-end deep reconstruction network recently[71,72]. Benefiting from the rapid progress of reconstruction algorithms, some groups have achieved snapshot spectral light field imaging[73,74].

In spite of the big progress, current snapshot spectral cameras stop at megapixel resolution and are not portable, confronted with severe challenges in terms of both system design and reconstruction algorithm development. Firstly, combining dispersion and spatial random coding is one typical way for CS based spectral camera[31], but scaling up such scheme to magnitude of ten-megapixel would result in bulky setup, expertise demanding engineering, or even go beyond the fabrication capability. Secondly, when the resolution exceeds mega pixels, the reconstruction algorithm is not readily available yet. Conventional iterative optimization might take months for reconstructing the spectrum of tens-of-millions pixels, while training an end-to-end deep neural network would take much longer time since spatially varying encoding pattern demands learning a large number of region-specific decoding networks. In spite that researchers are working on CS reconstruction of large scale data, such as plugin-and-play deep network[70] or fast adaption via meta learning[75,76], the performance is still not competitive to convex optimization and E2E deep network[77]. Overall, a lightweight design at tens-mega pixel scale leaves us two grand issues: First, how to fabricate a lightweight optical element to encode the large scale spatio-spectral data cube and conduct pixel-wise encoding without bulky relay optics? Second, how to computationally decode the multi-spectral data with both high performance and efficiency?

In this paper, we propose a tens-of-megapixel handheld snapshot multi-spectral camera (THETA), which combines compact setup design and algorithm development to push the spatial resolution of snapshot spectral imaging beyond tens of megapixels. In terms of hardware engineering, we design an imaging setup to produce a thin film with structured pattern performing multiplexed wavelength-dependent encoding, and directly mount the film onto the bare sensor of a high-resolution camera, forming a lightweight acquisition system.

Algorithmically, we perform a deep neural network based reconstruction as shown in Supplementary Fig. 5, with low training and inference cost while maintaining high performance. As a compact snapshot spectral camera capable of covering large scale nature scenes at fine details and with low lost, THETA is of advantageous performance and holds great promise in applications. we demonstrate our much higher throughput than current multi-spectral methods with proof-of-concept real applications—large-scale crop identification and health inspection, real-time monitoring of water pollution. Our method is expected to be applied on low capacity platforms or hand-held devices in the future, and hold great promise in agriculture, environmental science, geography, etc.

## Results
### The setup
The THETA setup is shown in Fig. 1a, similar to a commercial industrial camera compatible with F mount lenses, with a thin film mask attached in front of the sensor for compressive spectrum encoding (see Methods for details). The compact and lightweight design facilitates hand-held photography on low capacity platforms such as drones and mobile robots.

A magnified look of the film mask under the microscope is shown in the left panel of Fig. 1b, covering the 29.8 × 22.4 mm sensor with 65 megapixel counts (IDG-6500-M-G-CXP6). From the zoomed-in area one can observe inter-channel pattern difference and repetitive structures. We produce the coding mask using a commercial film camera (Mamiya RB67) and Fujichrome PROVIA 100F color positive film with exquisite grain, with the light path shown in Supplementary Fig. 3. Specifically, the mask is fabricated by superimposing a series of wavelength-dependent binary patterns onto the blank film, via shooting a photo-etched binary pattern (-18 μm resolution, quartz glass substrate) through a dispersive element that shifts the images of different wavelengths by varying amounts before entering the film camera. The disperser is fulfilled by a piece of planar glass (Schott N-SF66) with a low Abbe number, i.e., large chromatic aberration. We set the disperser at an acute angle with respect to the optical axis and place it either in front of the lens or between the lens and the sensor to conduct wavelength-dependent shifting. The snapshot is captured under broadband illumination (CME-303 fiber-coupled xenon light source, 300–2500 nm).

Denoting the transmission of the photo-etched binary pattern as $\mathbf{M}$, the spectra of the broadband environmental illumination and the blank film, respectively as $\mathbf{I}_0(\lambda)$ and $\mathbf{t}_{film}(\lambda)$, following the geometrical optics detailed in Supplementary Fig. 1, the transmission spectrum of the film mask can be quantitatively derived as

$$\mathbf{C}(x, y, \lambda) = \mathbf{M}\left(x - \Delta_x(x_0, \lambda), y - \Delta_y(y_0, \lambda)\right)\mathbf{I}_0(\lambda)\mathbf{t}_{film}(\lambda). \quad (1)$$

Here $\Delta_x(x_0, \lambda)$ and $\Delta_y(y_0, \lambda)$ are the wavelength-dependent lateral displacement introduced by the dispersive element, satisfying that

$$x_0 + \Delta_x(x_0, \lambda) = x \quad (2)$$

$$y_0 + \Delta_y(y_0, \lambda) = y. \quad (3)$$

In experiment, we discretize the integral into 12 spectral intervals and the set of $\{\mathbf{C}(x, y, \lambda)\}$ can be sequentially acquired by using the corresponding narrow-band spectral filters.

To circumvent the big challenge of adhering the film directly onto the bare sensor and avoid using bulky relay lenses with such large a space-bandwidth product (SBP), we propose to package a fiber optic plate (FOP, 9 mm thickness) onto the bare sensor and attach the film mask on its front face for a point-to-point transmission at high quality. The cross-section illustration of the FOP is shown on the right panel of

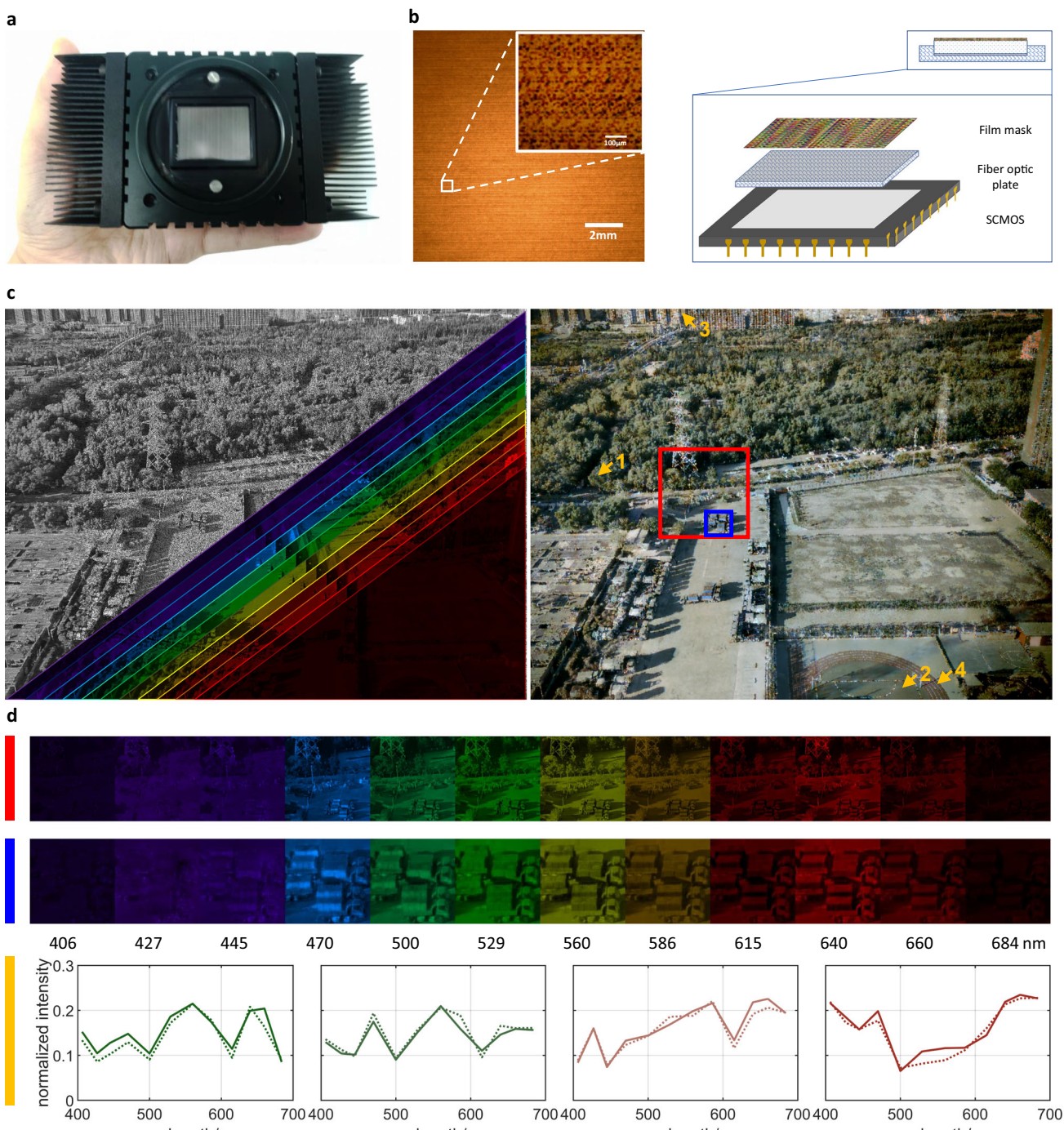

**Fig. 1 | A schematic diagram and result of THETA. a** The picture of THETA. **b** A local image of the proposed mask and the cross-sectional figure of the proposed system, where the zoomed-in view of the mask is acquired under a 40 × microscope. **c** The encoded measurement and multi-spectral images (left), and synthesized RGB view (right). **d** The multi-spectral reconstruction in the visible range of the region marked by the red and blue boxes in (**c**), respectively, colored by the RGB value of the corresponding wavelength, and the comparison between the retrieved spectrum (solid line) and the ground truth (dotted line) of four arrow-marked regions.

Fig. 1b, and one can refer to Supplementary Fig. 4 for the detailed packaging structure and the influence of FOP. Such a design also facilitates changing the film easily. During photography, we reserve a thread of about 9 mm between the lens and the camera to let the scene projected exactly on the front face of the FOP.

### Encoding scheme and decoding algorithm
After spectrum-dependent coding by $\mathbf{C}(x, y, \lambda)$, the intensity distribution of the encoded measurement can be derived in the integral form within the range of the film's response spectrum $[\lambda_1, \lambda_2]$ as

$$\mathbf{I}(x,y) = \int_{\lambda_1}^{\lambda_2} \mathbf{l}_1(\lambda) \cdot \mathbf{S}(x,y,\lambda)\mathbf{t}_{\text{FOP}}(\lambda)\mathbf{t}_{\text{rec}}(\lambda) \cdot \mathbf{C}(x,y,\lambda)d\lambda. \quad (4)$$

Here $\mathbf{S}(x, y, \lambda)$ is the spectrum of the target scene, $\mathbf{l}_1(\lambda)$ and $\mathbf{t}_{\text{FOP}}(\lambda)$ are the spectral profiles of the illumination and FOP's transmission, and $\mathbf{t}_{\text{rec}}(\lambda)$ denotes the spectral transmission of an additional customized filter mounted on the camera lens for

compensating the imbalanced transmission of the blank film with a reciprocal curve.

For efficient decoding of multi-spectral data at ten-megapixel scale, we propose to use an end-to-end deep neural network for reconstruction and adopt a coarse-to-fine strategy for fast model training. Since we use periodically repetitive pattern and the spectrum-dependent shifting is generally consistent across the film, the final encoding pattern is approximately structured, which is mathematically validated in Supplementary Fig. 2. There are around 1000 repetitive blocks on the film, among which nearby blocks are highly similar and there exists non-negligible difference among far apart ones, we group the blocks into 12 groups and learn their respective reconstruction networks. To further raise the training efficiency, we build a base model working for all the groups but with low reconstruction quality, and then adapt them to different groups fast. For each group, we employ a recently proposed network with state-of-the-art performance, designed with a sparse Transformer[77], with the network structure demonstrated in Supplementary Fig. 5 and corresponding descriptions of implementation details.

## System performance

In order to demonstrate the potential of THETA on depicting the spectrum of large field of view (FOV), we use a wide angle lens (BV-L1020) for an overhead photography of an urban landscape. The encoded measurement is displayed in the top-left corner of the left panel in Fig. 1c. Following the proposed pipeline we perform multi-spectral reconstruction to output the spatio-spectral data cube, as shown on the right-bottom half in an overlaid view. The reflection spectrum of natural objects is characterized by their intensities at different wavelengths, which can be accurately quantified by THETA. For an intuitive visualization of the target scene and displaying the advantageous of multi-spectral imaging over RGB counterpart, we synthesize a RGB image by accumulating the multi-spectral reconstruction result with the sensor's response curves, as shown in the right panel of Fig. 1c.

The zoomed-in multi-spectral images of two local regions of interest (ROIs), highlighted in Fig. 1c are displayed in top two rows of Fig. 1d. Benefiting from the high spatial resolution, the local details can be imaged quite well. To further demonstrate our capability of precise spectrum characterization, we remove the encoding film and place a set of narrow-band filters in front of the lens to record the ground truth spatio-spectral data cube. The spectral profiles of the regions marked by the four yellow arrowheads in Fig. 1c are plotted at the bottom of Fig. 1d. Here, the solid lines are the reconstructed spectrum defined by the average spectral profile, and dotted lines depict the ground truth. Although the four selected positions have the same or similar color in pairs in RGB image, the proposed approach is able to quantitatively illustrate their spectral differences, which is quite useful for applications demanding material identification among objects with close colors.

## Calibration of spatial and spectral resolution

In order to quantitatively test the spatial and spectral resolution of THETA, we use our setup to capture the ISO12233 Resolution Chart 2000 lines, on which four 2-inch circular band-pass filters and one broadband high-pass filter are placed over different line groups. The central wavelengths of narrow-band filters are 440, 550, 590, and 615 nm, respectively. A primary lens with 20 mm focal length is used and the image of 40 × 71 cm resolution chart covers 29.9 × 22.4 mm FOV of the sensor. After calibrating the transmission spectrum of the coding mask, we apply our reconstruction algorithm to retrieve the spatio-spectral data cube, with 7000 × 9344-pixel lateral resolution and 12 spectral channels. The encoded measurement and RGB image synthesized from the reconstructed spectral data cube and sensor's RGB response curves are shown in Fig. 2a. The multi-spectral images of

the region highlighted with orange box are displayed in Fig. 2b, where the two of the placed spectral filters with adjacent wavebands are partially included, demonstrating the precise spectral discrimination of THETA.

For quantitative evaluation, we firstly capture the ground truth spatio-spectral data cube by removing the encoding film and sequentially placing narrow-band filters in front of the camera lens. As for the spatial resolution, a comparison between the reconstructed and ground-truth intensity profiles of four line groups on the resolution target are plotted in Fig. 2c. Under the Rayleigh resolution criterion, we can derive that the average spatial resolution is about 2.16 pixels, i.e., 6.93 μm. To calibrate the spectral resolution, we use the color checker area at the bottom of the resolution target to characterize the spectral accuracy of the reconstruction algorithm. For visual comparison, we overlay the RGB image of the color checkers synthesized from their true spectral profiles onto Fig. 2a, above the reconstructed counterpart. Quantitatively, the profiles of three color squares are shown in Fig. 2d, in parallel with the ground truth plotted in dotted lines. One can see that the reconstructed profiles is quite close to the true value. Further, the spectral resolution can be quantitatively depicted by the multi-spectral imaging of the two narrow-band filters located within the orange highlighting box in Fig. 2a, which are with adjacent transmission wavebands. As shown in Fig. 2e, the same peak positions can be found in both the retrieved and ground-truth intensity of the two spectral filters, and can be easily distinguishable from each other. As the central wavelengths of the two filters are 590 nm and 615 nm, respectively, the spectral resolution of our method can achieve ~25 nm. Above resolution testing demonstrate that we can achieve snapshot spectral imaging at full sensor resolution and with high spectrum fidelity.

## Application 1: large area crop monitoring

To demonstrate the usability of THETA for outdoor scenes and the feasibility of applying for precision agriculture, we conduct overhead acquisition on a gridded farmland with nine different types of crops. Here a small-scale host is utilized to form a portable system that has been proven to be adaptable to most acquisition requirements and environments. As a portable setup, we can easily adjust its position and pose to flexibly balance the fineness of imaging on the crops and the field of view. Furthermore, when deployed on a mobile vision platform such as a drone, the video-rate acquisition allows efficient scanning of large fields with fine spatial and spectral scales. Specifically, the drone is only required to navigate stably according to the planned route and conduct full-frame capture for multi-spectral reconstruction. Neither is it necessary to hover at specific positions to capture multiple snapshots, which is time-consuming, nor is the subsequent registration and deblurring operations, thus reducing the time cost and complexity, while simplifying the post-processing process.

The coded snapshot, the RGB images respectively synthesized from the ground-truth and reconstructed multi-spectral data cube are shown in Fig. 3a, from left-top to right-bottom. From the RGB image, one can see that some different crops are of similar appearance and indistinguishable from each other. Then we conduct clustering on the reconstructed 12-channel multi-spectral data volume. Specifically, we firstly adopt K-means algorithm to divide the multi-spectral data into 33 clusters and then introduce spatial coordinates to serve as additional dimension for a secondary K-means clustering. The cascaded two-stage clustering divides the expanded data into 10 groups, as shown in the legend of Fig. 3b. The results show that we can reliably discriminate different crops of similar RGB intensities and even the subtle difference of the same crop. The enlarged differentiability among pixels validates the significance of multi-spectral imaging in crop monitoring: In case of widely adopted grid management in modern agriculture cultivating different species of crops in their

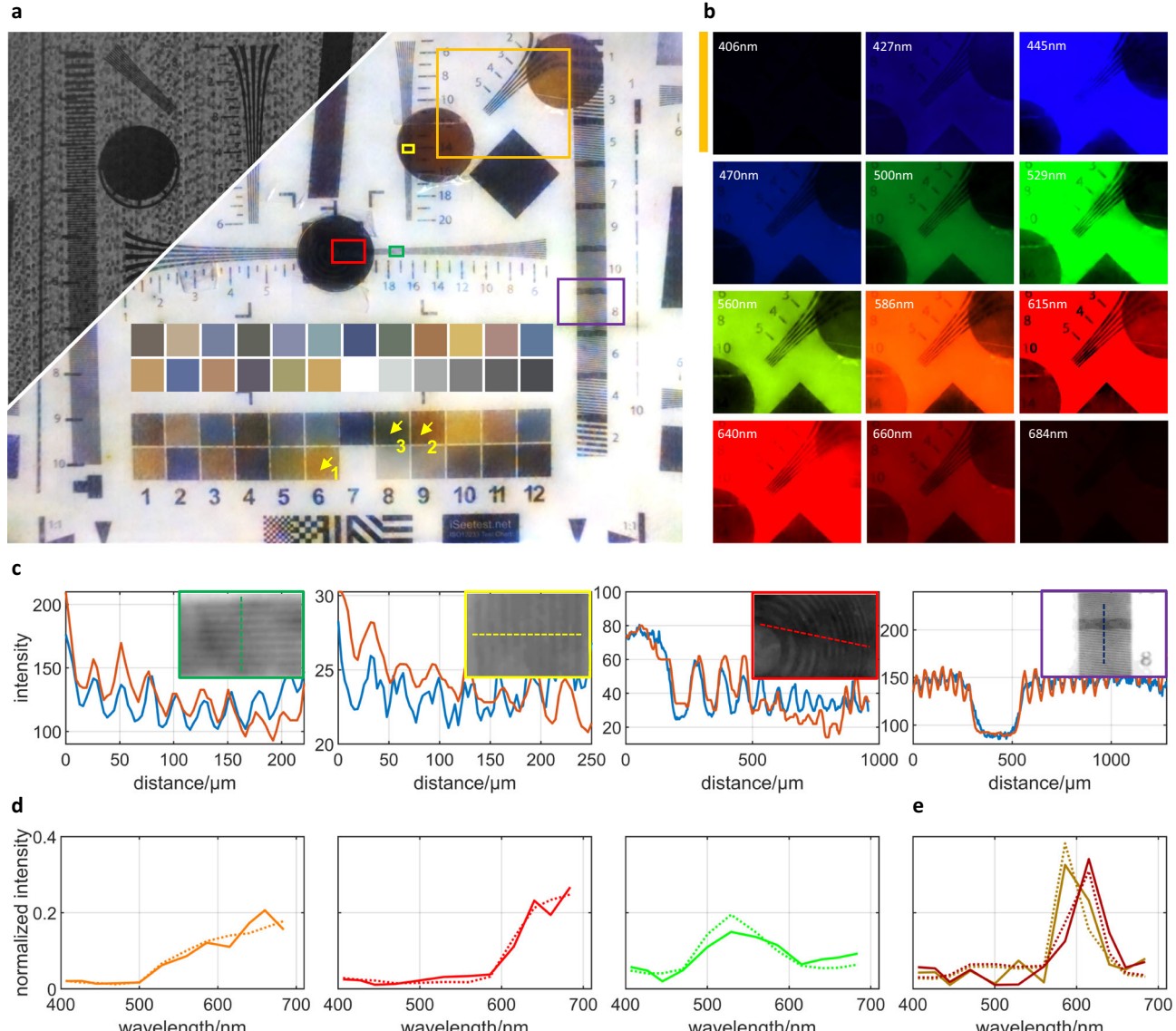

**Fig. 2 | The spatial and spectral resolution. a** The encoded measurement and the synthesized RGB image corresponding to the reconstructed multi-spectral data of the 2000-line resolution test target. The upper color checker area presents the ground truth of the color checker area under the response function of the adopted sensor. **b** Multi-spectral reconstruction result of the region marked in orange box in (**a**), colored by the RGB value of the corresponding wavelength. **c** The spatial line profiles for four regions on the resolution target, highlighted in green, yellow, red and purple boxes in (**a**), respectively. The orange lines represent the reconstructed results, and the blue lines the ground truth. **d** The spectral profiles for three spatial points marked by the yellow arrows in the color checker area in (**a**), plotted in orange, red and green lines, respectively. **e** The average spectral profiles for the two narrow-band spectral filters with adjacent transmission wavebands in the orange box in (**a**), marked in brown and red, respectively. In both (**d**) and (**e**), the solid lines represent the reconstructed results, and the dotted lines the ground truth.

respective regions, it is difficult to observe significant differences in large-scale RGB images at the early stage of plant diseases, while one can discriminate by clustering the spectral profiles to find the developmental anomalies in a wide range of similar crops and aid timely treatment.

Besides benefiting from the high throughput, the tens-of-megapixel resolution provides precise discrimination capability for individual plants with different health conditions, such as the cluster different from the surrounding plants shown in the green box in Fig. 3b. One can zoom in to conduct monitoring at finer scales. For example, the intensity of 12 spectral channels of four zoomed-in regions, respectively with four different types of crops, are provided in Fig. 3c. In summary, THETA has great potential for large-scale agriculture due to its compactness, full-image capture capability, and high data throughput, as it can distinguish different species, developmental stages, and health states, etc., based on spectral profiles, and it's

expected to be used on a continuously navigating drone for multi-spectral surveys with video rate.

**Application 2: real-time large-FoV water pollution monitoring**
We further validate the potential of the lightweight spectral camera for real-time observation. Considering that dynamic detection of sewage discharges is highly demanded for water health monitoring and of great significance to environmental protection, we build a miniature landscape of rugged terrain to simulate upstream sewage discharges and conduct overhead multi-spectral videography using THETA. Experimentally, we sequentially discharge two different pollutants, with similar RGB colors but different chemical substances, from the upstream water inlet. Figure 4a shows the encoded measurement and the RGB image synthesized from the 12 reconstructed channels at 99.6 seconds, while Fig. 4b shows the set of narrow-band images separately. The reconstructed video further shows the diffusion

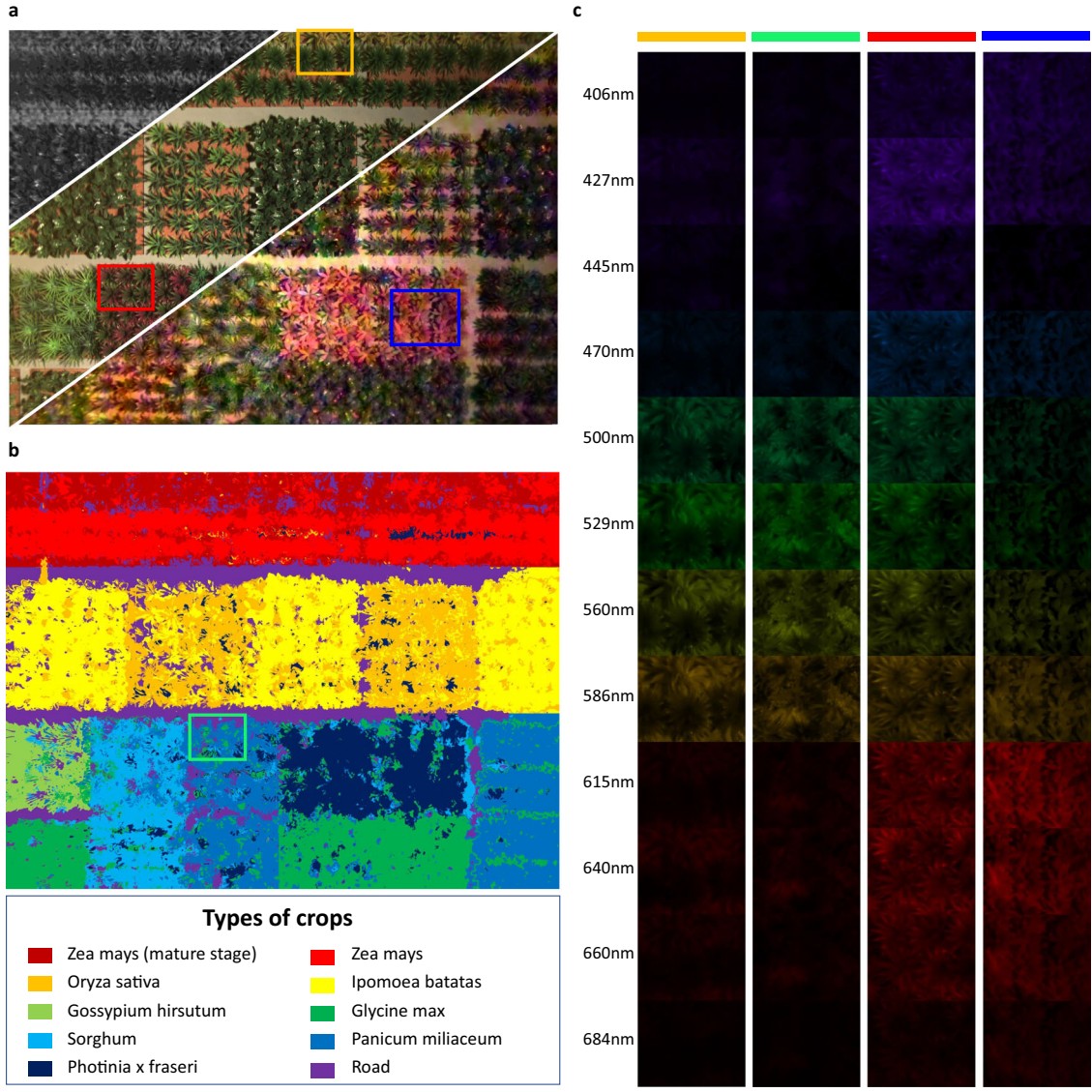

**Fig. 3 | Multi-spectral reconstruction of gridded farmland.** The encoded measurement, the RGB image from ground-truth and reconstructed spatio-spectral data volume. Top left: encoded measurement, middle: ground truth RGB image, bottom right: synthesized RGB image from the reconstructed result. **b** K-means segmentation result of the reconstructed multi-spectral data, showing the inherent spectral distinction of the 9 types of crops and the field path, where the green box marks the individual plants with different health conditions from their congener. **c** Multi-spectral image of the four regions marked in (**a**) and (**b**).

process of the two different colors in the water, as shown in Supplementary Movie 1.

In the upper row of Fig. 4c, we show six synthesized RGB frames of the whole field of view, and the zoomed-in view of the highlighted region in (a) is shown in the upper row of Fig. 4d. The spreading boundary and the increasing concentration cannot be observed directly. According to the spectral profiles of candidate pollutants, the multi-spectral data can be simply decomposed into components telling the location and concentration of multiple sewage types (see the lower row in Fig. 4c, d, in which we use two pseudo colors labeling two pollutants and the intensity characterizing their concentration). Please refer to Supplementary Movie 2 displaying the dynamic distribution of two pollutants. From the pseudo-color visualization of the sewage components and their diffusion, we can distinguish two types of sewage, quantitatively measure their concentration, and clearly infer their respective diffusion directions as well. Specifically, the sewage discharged earlier comes from the right side of ROI in Fig. 4d, while the later one comes from the bottom. The identification of diffusion amount, rate and source from the retrieved multi-spectral data cube is

of great help to the successive pollution traceability and treatment. Moreover, by integrating with a portable host, THETA can be easily mounted on a low-capacity drone for environmental monitoring over a large area with high precision.

## Discussion

In this paper, we propose a high-resolution handheld snapshot spectral camera (THETA), bypassing the grand challenges when encoded snapshot spectral imaging goes beyond megapixels, such as bulky setup, high cost, expertise demanding engineering, ultra-long inference time, etc. For compact encoded recording, we design an approach generating a thin planar spatio-spectral encoding element—a color film, and mount it onto the camera sensor for wavelength-dependent random coding without complex relay optics. In order for efficient decoding, the pattern on the film is designed to be approximately periodic. Such structured pattern facilitates efficient learning of decoding deep neural networks, via building a base model and performing fast adaption. Such property is specially important for tens-of-megapixel level reconstruction. To validate the proposed

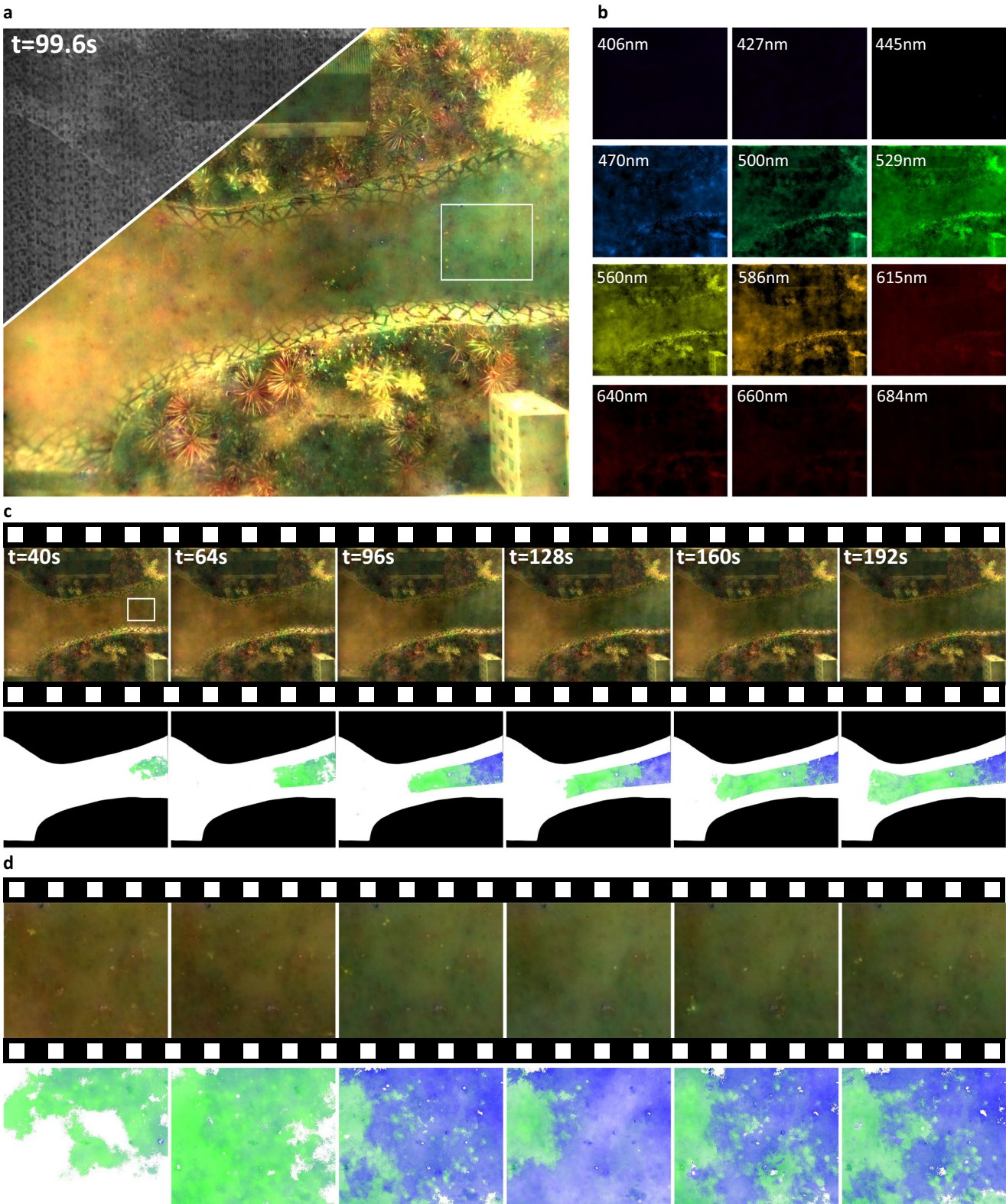

**Fig. 4 | Multi-spectral imaging of the diffusion process of sewage in a miniature landscape of a river. a** The encoded measurement (upper left corner) and RGB synthesized result at $t = 99.6$ s. **b** Multi-spectral reconstruction result at $t = 99.6$s, colored by the RGB value of the corresponding wavelength. **c** RGB synthesized time-lapse images and the corresponding heatmap generated from the multi-spectral data, showing the process of sewage diffusion. The brightness of the green and blue colors represent the concentration of the two types of sewage. **d** RGB synthesized time-lapse images and the corresponding heatmap for the region in white box in (**a**).

approach, we build a prototype achieving full resolution (65-megapixel spatial resolution) 12-band multi-spectral snapshot videography, which holds promise in capturing the spatio-spectral data cube over a large field of view, at high resolution and with spectrum fidelity.

## Advantages and wide applications

The core technical contributions are the multiplexed spectrum-dependent encoding film and the compact packaging technique. The proposed design contributes to the advantageous features of THETA: First, from the perspective of imaging mechanism, the film mitigates the expertise demanding high-precision optical engineering and expensive optical modulation components such as spatial light modulator (SLM) and disperser. The film is thin, light, cheap, open for structured pattern customization, and can go beyond the resolution upper limit due to physical constraints. Second, the packaging via FOP, leads to a lightweight imaging system that is significantly smaller in size and weight than existing hyperspectral imaging systems, which need bulky relay optics for pixel-wise encoding. The compact packaging and the customized encoding film together lay the foundation for the high throughput lightweight implementation. The size and weight can be further reduced if there exist new ways coupling the film directly onto the sensor plane. Third, in view of fabrication, the film surprisingly can be produced via film photography and simulated explicitly following geometrical optics. Moreover, it can be produced in batch with a commercial film camera under the same setting. In other words, the film can be manufactured via mass production.

As the imaging result of a highly complex natural scene in Supplementary Fig. 6 shows, the achieved resolution of 65 megapixels is guaranteed by both the hardware setup and the algorithm used, and does not merely serve as an interpolant to a conventional imaging device with much lower resolution. On the one hand, the proper system settings for film production cause both pixel-level encoding and pixel-level code shifting between adjacent spectral channels. Therefore, most of the effective information can be reserved by reconstructing statistically redundant natural scenes according to the principle of compressive sensing. On the other hand, the used transformer-based network is of well-designed architecture and has been trained on spectral data of various natural scenes, achieving performance superior to convex optimization. Furthermore, the proposed efficient reconstruction is highly scalable and can be applied to images with higher resolution. Although a higher resolution sensor with a larger sensory region may reduce the spatial consistency of the lateral displacement, the training time will not increase a lot since the coarse-to-fine adaption is fast. In terms of inference, the increased resolution only leads to linear growth in running time or graphics memory in parallel calculation.

As a general multi-spectral imaging approach, THETA can easily equip a commercial gray scale sensor with spectral imaging capability with only slight modifications, i.e., attaching a specific film with tailored resolution and size onto the sensor. For example, miniaturized spectroscopic cameras can be build on small commercial compact cameras. Likewise, the approach is equivalently compatible with sensors of higher spatial resolution, in spite of the challenges of developing higher efficiency reconstruction algorithm to match the pixel resolution.

Due to its high throughput and lightweight design, THETA has a wider range of applications besides smart agriculture and pollution monitoring demonstrated in this article. We expect that the camera can be mounted on different platforms and play roles in various fields such as geography, bio-medics, oceanology, etc.

## Future work

The encoding scheme is also applicable for microscopy, effectively discriminating different fluorescence excitations. We believe that this method can obtain broad applications in long-distance, wide-field observation and multi-color fluorescence imaging of small molecules. For example, in high-throughput gene sequencing and detection, our method is expected to effectively distinguish fluorescence excitations of different sequences, where spectral resolution plays an important role[78]. In monitoring and controlling of biological/chemical dynamics[79–81] the single-shot encoding scheme we applied could effectively improve the time resolution of imaging and help reveal the principle of microscopic phenomena. To integrate our film mask into a microscope setup for scientific observations, we have begun to explore the modification and packaging of high-sensitivity scientific sensors, which can be quite expensive and differs to some extent from current industrial design. Besides, the implementation of FOP in commercial microscopy systems can result in a mismatch in the length of the mount, requiring a custom mount design.

Extending to lensless microscopy for high throughput on-chip analysis is a topic worth further investigation, as lensless spectral imaging has already been studied and demonstrated advantageous in the size of the field-of-view, cost-effectiveness, and portability.

Similar to other imaging tasks with snapshot compression, altered noise levels or rich textures can lead to degradation of the performance and spatial resolution of the reconstruction. The former usually occurs when the noise in the measurement or calibrated masks is different from that in the model training. However, this can be mitigated by estimating the noise[82,83] and fine-tuning the network with data overlaid with the appropriate noise level. It has also been shown that increasing the number of spectra-aware hashing attention blocks makes the model perform better in the original paper of the adopted network[77]. The latter requires a shorter working distance to achieve effective coding by zooming in on the details, at the expense of the field of view (FOV). Another option is to select a sensor with a finer pixel size to achieve a higher resolution. Another factor worth mentioning is that in video capture, slight image distortion can occur due to the sensor's on-chip recording loss, resulting in degraded performance compared to static scenes. Using more advanced sensors can address this issue and further enhance the use of THETA in highly dynamic scenes. Recall that the experimental setup for film fabrication should be adapted to the modified sensor according to the geometric principles outlined in the "Film making" section of the Supplementary Information.

The number of spectral channels determined by the minimum wavelength interval with distinctive coding masks in our proposed method does not appear to be as large as in other CASSI-type implementations, but can be further improved by optimizing the capability of the dispersion components, the choice of the film camera, and the pixel size of both the lithography mask and the sensor. Specifically, the displacement between adjacent spectral channel masks generated by dispersion must be at least as large as the pixel spacing of the sensor to enable effective spectral encoding. Theoretically, the displacement depends on the system setup for film fabrication, as described in the Methods and Supplementary Information: A smaller distance from the aperture to the lithography mask, a lower Abbe number, and a larger thickness of the dispersive element result in a larger lateral shift. However, there is a trade-off among sufficient lateral shift, the effectiveness of encoding, and film quality. First, a smaller object distance means a larger system magnification, which in turn means that the size of each mask cell recorded on the film significantly exceeds the pixel size of the sensor due to the limited precision of lithographic fabrication, making pixel-wise encoding difficult. An object distance less than the minimum working distance of the film camera can also cause image distortion. An optimized object distance can be achieved by exploring other types of film cameras and other manufacturing methods for binary masks with higher precision. Second, increasing the thickness of the dispersion element simultaneously increases the lateral and axial distances between the imaging results of the different spectral channels. While the former benefits spectral encoding, the

latter can lead to defocusing of the image. In addition, internal dispersion in thick media affects image quality to some degree. Selecting dispersion elements with better performance can overcome this limitation. Third, the lower the Abbe number of the dispersion element, the stronger the dispersion, but this presents the same challenge as increasing the thickness for axial displacement. In summary, the film fabrication process is the key factor on which the number of spectral channels and the spectral resolution of our proposed system depend. Using the existing pixel size (3.2 mm) as the lower limit of the shift, we properly set the experimental parameters (see the "Film making" section in the Supplementary Information for details) and obtained information from multi-spectral channels with a number of 12, which is expected to be optimized by modified hardware.

For the film itself, despite its low cost and portability, there also exist some inherent limitations. On the one hand, the film absorbance would result in reduced luminous flux and suffer from noise under dark illumination. On the other hand, the spatial resolution of the film is fundamentally determined by its granularity and might be of insufficient modulation capabilities for smaller pixels. We are looking for new materials that can record the patterns in finer detail and serve as a better spectrum encoding element.

Besides, applying the proposed method for some task-specific analysis other than reconstructing the whole spectrum may be expected to further simplify the system, raise the running efficiency or increase the SBP.

## Methods

### Fabrication of the encoding film
In order to achieve spatio-spectral random encoding applicable for tens-of-megapixel image systems in a compact form, we propose to generate a film with color-dependent random transmission patterns. The fabrication is conducted by introducing spectrally distinctive shifting during film photography of a binary random pattern. Specifically, we use a commercial film camera and color-positive film to shoot a high-resolution binary pattern etched on a plate glass by lithography through an oblique planar diserperser, which shifts the different spectral components of the etched random pattern by different offsets. The lithographic pattern is designed to be array repetitive, and the shifting is approximately uniform across the field of view, so the final recorded spectral code is of a globally structured layout, which facilitates efficient training of reconstruction models corresponding to different regions.

In implementation, we use Mamiya RB67 film camera and Fujichrome PROVIA 100F color-positive film for recording. The grid size of the chrome mask is 3.84 μm, and the grid is arranged by periodically repeating the 256 × 256 random pattern. The magnification of the film camera imaging system is about 0.83×. The dispersion is generated by a piece of Schott N-SF66 glass (thickness 7 mm) placed in front of the lens at a certain angle (around 40°). During acquisition, we illuminate the lithographic patterns with a fiber-coupled xenon light source (CME-303 solar simulator manufactured by Microenerg), which has a relatively uniform spectrum in the visible spectrum. Besides, we further design a conjugated two-arm setup—film and digital for high precision adjustment of optical path, with the output of digital camera for real time monitoring of the expected quality of film pattern, as shown in the diagram in Supplementary Fig. 3.

It can be verified (see Supplementary Fig. 2 and corresponding descriptions) that the shifting is spectrally distinctive and the spectrum-specific displacements are roughly consistent across the camera sensor, which ensures the spatial periodicity of the fabricated film mask. The conclusions are further illustrated by the correlation heat-maps of masks at different wavelengths and positions, respectively, demonstrating that our approach can generate high spatial resolution mask with high spectral encoding capability and spatial periodicity.

### Compact integration of the coding mask
The proposed multi-spectral imaging system THETA is composed of the IDG-6500-M-G-CXP6 camera with 65 million pixels, mounted with an M58 to F-Mount adapter to be compatible with commercial F-Mount lenses. Since a lightweight relay lens with the space-bandwidth product covering 65 million pixels is rarely commercially available, and directly attaching the film on a bare sensor is expertise-demanding, risky, and experimentally irreversible once packaged, we propose to use an FOP to transmit the coding mask onto the sensor.

In THETA, we use a 31 × 23 × 9 mm FOP to relay the film mask onto the sensor plane. For easier engineering, we use a sensor with removable cover glass to facilitate mounting the FOP on the bare sensor surface in a dust-free environment. Then the film mask is fixed on the outer surface of the FOP with a customized piece of cover glass. The experiment in Supplementary Fig. 4 proves that FOP copies the imaging of the front surface to the rear counterpart with low frequency loss and high transmittance.

When using the customized sensor for encoded multi-spectral imaging, one can simply fine-tune the image plane of the target scene onto FOP's front surface, and then the scene is encoded by the film mask and its encoded measurement is relayed and recorded by the sensor.

### System calibration
A series of band-pass filters (Central wavelength/Guaranteed minimum bandwidth (nm): 406/15, 427/10, 445/20, 472/22, 500/24, 529/24, 560/25, 586/20, 615/24, 640/14, 660/13, 686/24) produced by Semrock are used to calibrate the masks at different wavelengths, and the 12 calibrated masks and coded acquisition are jointly fed to the reconstruction network to retrieve the multi-spectral images. The intrinsic transmittance of the system (including film and FOP) at the imaging wavelengths is obtained by capturing the whiteboard at the same exposure setting.

### Encoded imaging model
The intensity of encoded measurement at coordinate $(x, y)$ is jointly determined by the spectrum of the light source, the spectrum at the corresponding position of the target scene $\mathbf{S}(x, y, \lambda)$, the intrinsic transmission spectrum of the FOP $\mathbf{t}_{\text{FOP}}(\lambda)$ and that of the film mask $\mathbf{C}(x, y, \lambda)$, i.e.,

$$\mathbf{I}(x,y) = \int_{\lambda_1}^{\lambda_2} \mathbf{I}_1(\lambda) \cdot \mathbf{S}(x,y,\lambda) \mathbf{t}_{\text{FOP}}(\lambda) \mathbf{t}_{\text{rec}}(\lambda) \cdot \mathbf{C}(x,y,\lambda) d\lambda, \quad (5)$$

where $[\lambda_1, \lambda_2]$ is the range of film's response spectrum, $\mathbf{I}_1(\lambda)$ is the spectrum of the illumination (after flat field correction) and $\mathbf{t}_{\text{rec}}(\lambda)$ is the "reciprocal" transmission curve of blank film to correct the inherent unbalanced spectral response of the film.

According to the geometrics of the light path for film mask fabrication (please see Supplementary Fig. 1), the transmission of wavelength $\lambda$ at position $(x, y)$ is equal to that of the binary mask at position $(x_0, y_0)$, with its offset $(\Delta_x(x_0, \lambda), \Delta_y(y_0, \lambda))$ being introduced by the dispersive element. Besides, the coding mask is also jointly determined by the spectrum transmission of film $\mathbf{t}_{\text{film}}$ and the illumination used for fabricating the mask $\mathbf{I}_0(\lambda)$, i.e.,

$$\mathbf{C}(x,y,\lambda) = \mathbf{M}\left(x - \Delta_x(x_0,\lambda), y - \Delta_y(y_0,\lambda)\right) \mathbf{I}_0(\lambda) \mathbf{t}_{\text{film}}(\lambda), \quad (6)$$

where $\mathbf{M_0}$ is the original binary mask, while $\Delta_x(\cdot)$ and $\Delta_y(\cdot)$ calculate the derived lateral displacement after inserting the disperser (see Supplementary Information).

### Multi-spectral reconstruction
It is non-trivial to decode tens-of-megapixel multi-spectral data with both high quality and efficiency. Benefiting from the structured layout

of the encoding film, we propose to use a state-of-the-art neural network that embeds spectral sparsity into transformer and train 12 networks in a coarse-to-fine manner[77] for block-wise reconstruction of the whole spatio-spectral data cube.

The network takes the snapshot encoded measurement and calibrated multi-spectral masks as input, and consists of two subsequent U-Net structured modules for decoding, with the network structure illustrated in Supplementary Fig. 5. The first module is a sparsity estimator, which depicts the regions with the sparsity of spectral information and produces a sparsity mask **M**. The second one filters the regions with dense spectral information according to the binarized sparsity mask, and guides the focus of the multi-head self-attention (MSA) mechanism[84]. Different from the ordinary MSA mechanism, the network uses hash mapping to divide the pixels in each selected region into buckets and apply multi-round MSA respectively, which limits the self attention within related regions and raises running efficiency largely (see Supplementary Information for details). The model training is conducted via minimizing following loss function

$$\mathcal{L} = \parallel \mathbf{S} - \mathbf{S}^* \parallel_2 + \lambda \cdot \parallel \mathbf{M} - \mathbf{M}^* \parallel_2. \qquad (7)$$

Here the first term penalizes large deviation from the true spectrum **S** and the estimation **S**$^*$ of the target scene; while the second term forces the estimated sparsity mask **M** to be close to its reference **M**$^*$, which demonstrates the distribution map of required attentions derived by the channel-wise average difference between **S** and **S**$^*$.

## Data availability
The source image and source data in the figures generated in this study have been deposited under DOI link[85]. The raw data generated in this study for the statistical plots in Figs. 1 and 2 are provided in the Source Data file. Source data are provided with this paper.

## Code availability
The code used in this study is available from the Zenodo repositories, respectively, for the adopted network[86] and the simulation of dispersion[87].

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

## Acknowledgements

This research is jointly supported by Ministry of Science and Technology of the People's Republic of China (Grant No. 2020AAA0108202 received by J.S.) and the National Natural Science Foundation of China (Grant Nos. 61931012 received by J.S., 62088102 received by Q.D.).

## Author contributions

J.S. and W.Z. conceived this project. Q.D. and J.S. supervised this research. J.S. and W.Z. designed the experiments and analysed the results. W.Z., K.D. and L.L. conducted the experiments. X.Y. supervised the operation of the decoding neural network. C.P. conducted the packaging of the system. All the authors participated in the writing of this paper.

## Competing interests

The authors declare no competing interests.
