## [Peer Review File · Nature Communications]

REVIEWER COMMENTS

Reviewer #1 (Remarks to the Author):

The authors have proposed and demonstrated a compact 12 channel 65 megapixel multispectral camera. The most notable feature of the design is its compact size, which makes simple drone-based data capture feasible and attractive. The pixel count is also very likely a record for multispectral imaging systems. The data estimation and analytical tools used on this camera are also state-of-the-art. I believe that the feasibility of such an imaging system will be of interest to a relatively broad community.

This is an engineering report, the results are presented well and are accurate, the image quality shown is impressive. The use of a film encoding layer and fiber optic channels substantially reduces the size per resolved pixel. The staring aspect of the design is perhaps the most compelling feature. Analysis of slowly changing scenes, as in crop analysis, do not require video-rate data acquisition. It would be interesting to discuss why full frame capture is needed in such applications or to consider applications such as chemical process control or transient target detection where full frame capture is required.

Reviewer #2 (Remarks to the Author):

This work addresses the important application of hyperspectral imaging returning lots of pixels to the user -- 65M in the implementation. While demonstrating little novelty in their implementation, the authors clearly did a lot of hard work and fine tuning to accomplish their results. The imager design is based on the CASSI principle of coded aperture with lots of engineering improvements. The algorithm is adopted from a recent ECCV paper showing how to use the transformer principle for hyperspectral imaging [Supp. Mat. ref #2] The authors should definitely be rewarded for their hard work by getting this paper published. My main concern is that the validations on calibrated targets may be inadequate. The authors should examine how their neural networks operate, and cite performance limits or caveats where their adopted operational principles and engineering fine-tunings are no longer valid: for example, levels of noise in the measurement; highly complex scenes that may violate the priors learnt by the neural network; etc. Ultimately, does the system return truly 65M degrees of freedom (pixels), or is it just an interpolant on a more traditional <1Mpixel imager without any guarantees on the information carried by the interpolated pixels?

Reviewer #3 (Remarks to the Author):

Manuscript describes an integrated snapshot spectral camera. The physical design is novel and interesting. The architecture is based on well know methods of spatial/spectral compressive snapshot cameras. The number of spectral channels seem small compared to other CASSI-type implementations. A description of the spectral resolution limits of the camera would be useful. In particular, what optical elements or measurement phenomena limits the spectral resolution. A more thorough description of the coding mask fabrication would be useful as well. In the review of prior work on tunable filters, the paper could cite Xi Wang, et. al "Compressive spectral imaging system based on liquid crystal tunable filter," Opt. Express 26, (2018), where compressive measurements are used in concert with liquid crystal tunable filters. Overall, the paper is a good contribution to the literature on spectral snapshot cameras, particularly from a fabrication perspective.

Response Letter for "Handheld Snapshot Multi-spectral Camera at Tens-of-Megapixel Resolution"

Responses to Reviewer #1's comments:

Overall Evaluation: The authors have proposed and demonstrated a compact 12 channel 65 megapixel multispectral camera. The most notable feature of the design is its compact size, which makes simple drone-based data capture feasible and attractive. The pixel count is also very likely a record for multispectral imaging systems. The data estimation and analytical tools used on this camera are also state-of-the-art. I believe that the feasibility of such an imaging system will be of interest to a relatively broad community.

This is an engineering report, the results are presented well and are accurate, the image quality shown is impressive. The use of a film encoding layer and fiber optic channels substantially reduces the size per resolved pixel. The staring aspect of the design is perhaps the most compelling feature.

Response: Many thanks for the positive comments, and for highlighting the advantages as well as the big application potentials of the proposed approach.

Q1: *Analysis of slowly changing scenes, as in crop analysis, do not require video-rate data acquisition. It would be interesting to discuss why full frame capture is needed in such applications or to consider applications such as chemical process control or transient target detection where full frame capture is required.*

A1: In fact, we agree that when used as a fixed camera monitoring slowly changing scenes, full frame capture is indeed unnecessary. However, we here are devoted to demonstrating a multi-spectral imaging paradigm---a mobile vision platform (e.g., drone) equipped with a high throughput lightweight multi-spectral camera to allow efficient scanning of large fields with fine spatial and spectral scales. To this end, full frame capture becomes necessary because conventional multi-shot spectral imaging approaches require the drone to hover at a position for a period to capture multiple snapshots, which is time-consuming, and the inevitable vehicle shaking would lead to additional misregistration and motion-blur issues. We have supplemented the discussion on the necessity of using full frame capture in the application section of the revised manuscript.

As for the applications in observing transient processes in the biological/chemical field, it is a part of our future direction, e.g., extending to high throughput gene sequencing via multi-spectral microscopy of gene chips with a similar setup. However, such applications involve modification and packaging of highly sensitive scientific sensors, which are with different structures from industrial cameras and are quite expensive, so related designs are still in progress. We also supplement this point in the discussion section of the revised manuscript.

Responses to Reviewer #2's comments:

This work addresses the important application of hyperspectral imaging returning lots of pixels to the user -- 65M in the implementation. While demonstrating little novelty in their implementation, the authors clearly did a lot of hard work and fine tuning to accomplish their results. The imager design is based on the CASSI principle of coded aperture with lots of engineering improvements. The algorithm is adopted from a recent ECCV paper showing how to use the transformer principle for hyperspectral imaging [Supp. Mat. ref #2] The authors should definitely be rewarded for their hard work by getting this paper published.

Q1: *My main concern is that the validations on calibrated targets may be inadequate. The authors should examine how their neural networks operate, and cite performance limits or caveats where their adopted operational principles and engineering fine-tunings are no longer valid: for example, levels of noise in the measurement; highly complex scenes that may violate the priors learnt by the neural network; etc. Ultimately, does the system return truly 65M degrees of freedom (pixels), or is it just an interpolant on a more traditional <1Mpixel imager without any guarantees on the information carried by the interpolated pixels?*

A1: Thanks for your suggestion. The proposed camera does bring the true information of 65 million pixels for the following two reasons. On the one hand, the device performs pixel-wise encoding since the masks on the fabricated film is of pixel-level shift between adjacent spectral channels, and the principle of compressive sensing tells that high fidelity reconstruction is feasible for statistically redundant nature scenes. On the other hand, the adopted neural network is of well-designed architecture and trained from spectral data of diverse nature scenes, and thus experimentally achieves state-of-the-art, high reconstruction performance. In Figure 1 in this letter and Supplementary Figure 6 in the revised supplementary material, we provide an additional visual demonstration of the multi-spectral imaging for a complex nature scene containing various growing plants, and the zoomed-in view on the texture features of leaf veins and petal edges is also displayed. As for the performance limit, we agree that the quality might degenerate for the circumstances where the noise in the measurement differs from that in model training, or the scenes are with highly complex details or rich texture. The former can be mitigated by estimating the noise and fine-tuning the network with data overlaid with corresponding noise level. Increasing the number of spectra-aware hashing attention blocks (i.e., changing the employed model from CST-S to CST-M or CST-L) is also proven to offer better performance on the acquired data as demonstrated in the original paper of the adopted network. The latter is a common issue for all the snapshot computational photography, and one can raise the spatial resolution via using a lens with longer focal length, capture from a nearer distance or alternatively select a sensor with finer pixel size, which would further magnify the details to reveal the redundancy and thus obtain high quality reconstruction. Notably the experimental setup of the film fabrication should be correspondingly adjusted for the altered sensor according to the principle discussed in the response to Reviewer #3. We have discussed these issues in the "Summary and Discussions" section of the revised version.

Figure 1. The multi-spectral reconstruction of a complex nature scene containing various growing plants. (a) The encoded measurement and multi-spectral images (left), and synthesized RGB view (right). (b) The multi-spectral reconstruction in the visible range of the region marked by the red and blue boxes in (a) respectively, colored by the RGB value of the corresponding wavelength.

Responses to Reviewer #3's comments:

Q1: *The number of spectral channels seem small compared to other CASSI-type implementations. A description of the spectral resolution limits of the camera would be useful. In particular, what optical elements or measurement phenomena limits the spectral resolution. A more thorough description of the coding mask fabrication would be useful as well.*

A1: Thank you for your positive comments and kind suggestions about our work.

a. The factors determining the spectral resolution of the camera.

The number of spectral channels is determined by the minimum wavelength interval with distinctive coding masks, calculated at long wavelength since the "Film making" section in the supplementary material tells that the longer wavelength is of smaller shift (masks with lower distinguishability). Firstly, the number of spectral channels is mainly limited by the system setup for film fabrication, specifically the object distance to the film camera and the specifications of the dispersion element, including the thickness, the Abbe number as well as the angle to the optical axis.

Secondly, the spectral resolution is also related to the size of sensor's pixel pitch, which should be smaller than the shift difference between the coding masks of adjacent spectral channels.

Thirdly, the ultimate resolution is from trade-offs between the above offset value and the sharpness of the wavelength dependent masks. (i) for the same film camera, a shorter object distance produces larger offset, but due to the limited precision of the lithography binary mask a too small object distance could result in over-magnification hampering pixel-wise encoding (i.e., the image of each mask cell exceeds that of the sensor unit). In addition, an object distance below the minimum

working distance of the film camera may lead to large distortion. (ii) thicker dispersion element generates larger offset but would degrade the quality of coding masks on the film due to internal scattering and increased axial displacement among channels. (iii) the angle between the dispersion element and the optical axis also exhibits a nonlinear relationship with the offset, which comes to peak around 40~45 degrees.

b. The settings in our implementation and calculation of the spectral resolution.

Dispersion elements: we have adopted a type of dispersion glass with the lowest Abbe number in Schott's product list, with a thickness of 7 mm to keep the variation of axial displacement within 1 mm and an angle with the optical axis of about 40 degrees.

Lithography mask and photography settings: The object distance is set to the minimum working distance of 300 mm, resulting in a magnification about 0.83. Therefore, we have designed a lithography mask with the unit size of 3.84 μm to present a 3.2 μm array on the film.

Camera sensor: We use a CMOS sensor with pixel pitch of 3.2 μm .

The calculation of the spectral resolution: According to our derivation, a lateral offset of about 3.2 μm can be generated on the image plane between the calibrated masks of the spectral channels with center wavelengths of 640 nm and 660 nm, respectively, then 12 spectral channels with intervals of 20~30 nm are selected for the experiment, which can be generated by easily available commercial filters without customization.

In sum, the number of spectral channels determined by the system setup for film fabrication and the sensor's pixel size, and we can use higher dispersive elements, or other types of film cameras, increase the lithography accuracy, or use a sensor with smaller pixel size to further improve the spectral resolution. We discuss the factors affecting the number of spectral channels (i.e., spectral resolution) in the Summary and Discussions section of the revised manuscript, and provided the numerical values and quantitative illustration of the film fabrication in the supplemental material.

Q2: *In the review of prior work on tunable filters, the paper could cite Xi Wang, et. al "Compressive spectral imaging system based on liquid crystal tunable filter," Opt. Express 26, (2018), where compressive measurements are used in concert with liquid crystal tunable filters. Overall, the paper is a good contribution to the literature on spectral snapshot cameras, particularly from a fabrication perspective.*

A2: Thanks for your suggestion. We apologize for missing this important work in our literature review. The referred paper by Wang et al. serves as a typical representative of compressive multispectral imaging using tunable filters, and is an alternative to CASSI or its variants. Although requiring multiple snapshots and inapplicable for highly dynamic observation, this design achieves a significant improvement over conventional LCTF-based spectrometer and offers multiple advantages, e.g., ease of use, high image quality, and light weight. We have cited and briefly introduced this work in the Introduction section of the revised manuscript.

REVIEWERS' COMMENTS

Reviewer #1 (Remarks to the Author):

In my opinion, the authors have addressed concerns raised in review and the manuscript should now be accepted.

Reviewer #2 (Remarks to the Author):

The responses to the reviewers' comments were adequate.

Reviewer #3 (Remarks to the Author):

The authors have adequately addressed all comments provided previously. The paper is a good contribution to the literature and it is recommended for publication.

Response Letter for "Handheld Snapshot Multi-spectral Camera at Tens-of-Megapixel Resolution"

Responses to Reviewer #1's comments:

In my opinion, the authors have addressed concerns raised in review and the manuscript should now be accepted.

Response: We would like to highly appreciate your valuable efforts reviewing our manuscript, and also express many thanks for the positive comments and suggestions on our proposed approach.

Responses to Reviewer #2's comments:

The responses to the reviewers' comments were adequate.

Response: We would like to highly appreciate your kind efforts reviewing our manuscript, and express many thanks for your valuable suggestions and positive feedback on our proposed approach.

Responses to Reviewer #3's comments:

The authors have adequately addressed all comments provided previously. The paper is a good contribution to the literature and it is recommended for publication.

Response: We highly appreciate for your kind efforts reviewing our manuscript, and thank you for your valuable suggestions and positive feedback on our proposed approach.